

# Biogeographic evidence supports the Old Amazon hypothesis for the formation of the Amazon fluvial system

Karen Méndez-Camacho[1], Omar Leon-Alvarado[1,2] and Daniel R. Miranda-Esquivel[1]

[1] Biology school, Universidad Industrial de Santander, Bucaramanga, Santander, Colombia
[2] Programa de Pós-Graduação em Biodiversidade Animal, Universidade Federal de Santa Maria, Santa Maria, Rio Grande do Sul, Brazil

## ABSTRACT

The Amazon has high biodiversity, which has been attributed to different geological events such as the formation of rivers. The Old and Young Amazon hypotheses have been proposed regarding the date of the formation of the Amazon basin. Different studies of historical biogeography support the Young Amazon model, however, most studies use secondary calibrations or are performed at the population level, preventing evaluation of a possible older formation of the Amazon basin. Here, we evaluated the fit of molecular phylogenetic and biogeographic data to previous models regarding the age of formation of the Amazon fluvial system. We reconstructed time-calibrated molecular phylogenies through Bayesian inference for six taxa belonging to Amphibia, Aves, Insecta and Mammalia, using both, nuclear and mitochondrial DNA sequence data and fossils as calibration points, and explored priors for both data sources. We detected the most plausible vicariant barriers for each phylogeny and performed an ancestral reconstruction analysis using areas bounded by major Amazonian rivers, and therefore, evaluated the effect of different dispersal rates over time based on geological and biogeographical information. The majority of the genes analyzed fit a relaxed clock model. The log normal distribution fits better and leads to more precise age estimations than the exponential distribution. The data suggested that the first dispersals to the Amazon basin occurred to Western Amazonia from 16.2–10.4 Ma, and the taxa covered most of the areas of the Amazon basin between 12.2–6.2 Ma. Additionally, regardless of the method, we obtained evidence for two rivers: Tocantins and Madeira, acting as vicariant barriers. Given the molecular and biogeographical analyses, we found that some taxa were fitted to the "Old Amazon" model.

## INTRODUCTION

The Amazon basin harbors a high biodiversity, which has been attributed to different geological events such as the formation of rivers in a scenario that was proposed by *Wallace (1852)* known as the "riverine barrier hypothesis". Wallace's idea was formalized by *Cracraft (1985)* who presented a regionalization based on bird distributions and the major Amazonian rivers. Over the years this regionalization has changed based on updated

Corresponding author
Daniel R. Miranda-Esquivel, dmiranda@uis.edu.co

information and methods (*Silva, Novaes & Oren, 2002*; *Borges & Silva, 2012*), providing evidence either supporting or rejecting the "riverine barrier hypothesis" (*e.g.*, *Hayes & Sewlal, 2007*; *Maldonado-Coelho et al., 2013*; *d'Horta et al., 2013*; *Oliveira, Vasconcelos & Santos, 2017*), and even reconsidering the rivers as vicariant barriers (*Pirani et al., 2019*).

Likewise, over the years geological studies based on sedimentation, mineralogy, lithology, paleoenvironmental conditions and fossils (*e.g.*, *Hoorn et al., 1995*; *Hoorn et al., 2010*; *Hoorn et al., 2017*; *Wesselingh et al., 2002*; *Wesselingh et al., 2006*; *Latrubesse et al., 2010*; *Antoine et al., 2013*; *Antoine et al., 2016*; *Horbe et al., 2013*; *Matos-Maravi et al., 2013*; *Rossetti et al., 2015*; *Jaramillo et al., 2017*) have been focused on the formation of the Amazonian basin, proposing two geological models: The "Old Amazon" and the "Young Amazon". In those studies the temporal range of the formation of the basin remains controversial, however it is clear the influence and importance of different geological events that shaped and contributed to the formation of the actual Amazon basin . The first important event was the gradual uplift of the Eastern Cordillera in the Central and Northern Andes, which caused the closure of the Western Andean Portal and created a watershed in the middle Miocene (∼16.1–12.4 Ma, *Hoorn, 1993*; *Hoorn et al., 2010*; *Hoorn et al., 2017*; *Parra et al., 2009*; *Antoine et al., 2013*; *Antoine et al., 2016*; *Jaramillo et al., 2017*; *Horton, 2018*) known as the Pebas mega-wetland system (hereafter, PMWS). This watershed was located in the Western Amazonia and was connected to the Caribbean Sea and bounded by the Purus Arch on the east (*Figueiredo et al., 2009*; *Figueiredo et al., 2010*). In the middle late Miocene, the PMWS disappeared (*Wesselingh et al., 2002*; *Wesselingh et al., 2006*; *Antoine et al., 2013*; *Antoine et al., 2016*; *Jaramillo et al., 2017*) due to the uplift of the Eastern Colombian Andes (∼15.0–3.0 Ma, *Mora et al., 2008*), and as consequence, the Vaupés Arch was drawn close to the Andes, causing the Orinoco-Amazonas separation event (*Olivares et al., 2013*; *Jaramillo et al., 2017*) and gradually reducing the water flux towards the Caribbean (*Lundberg et al., 1998*). Finally, due to the subsidence of Purus Arch, the Amazonas river changed its primary westward flow into the Pacific Ocean to an eastward flow discharging into the Atlantic Ocean (*Dobson, Dickens & Rea, 2001*; *Figueiredo et al., 2009*; *Figueiredo et al., 2010*; *Nogueira, Silveira & Guimarães, 2013*; *Rossetti et al., 2015*; *Hoorn et al., 2017*; *Van Soelen et al., 2017*). Nevertheless, the dates when the Amazon River changes its flow and onset of the Amazon basin are still in debate, and are the main differences between the two geological models proposed. The "Old Amazon" model suggests that the fluvial system was established during the middle late Miocene (∼10.0–7.0 Ma, *Hoorn et al., 1995*; *Hoorn et al., 2010*; *Hoorn et al., 2017*; *Figueiredo et al., 2009*), while some authors consider an upper limit of 10.5 Ma (*Figueiredo et al., 2010*), or even 14.0 Ma (*Shephard et al., 2010*). On the other hand, the "Young Amazon" model covers a wider temporal range, with studies agreeing that the basin was completely formed during the Pliocene-Pleistocene (∼2.5 Ma, *Campbell, Frailey & Romero-Pittman, 2006*; *Latrubesse et al., 2010*; *Rossetti et al., 2015*). Both models have been evaluated using biogeographical qualitative (*e.g.*, *Ribas et al., 2011*; *Fernandes, Wink & Aleixo, 2012*) and quantitative (*Quintero et al., 2015*) approaches, which suggested that the Amazon basin was formed in the Pliocene-Pleistocene, agreeing with the "Young Amazon" geological model. These biogeographical studies also have found how some amazonian rivers influenced the evolutionary history of the species

(*e.g.*, *Fernandes, Wink & Aleixo, 2012*; *Alfaro et al., 2015*), and established regions that matched with some main Amazonian rivers (*Godinho & da Silva, 2018*; *Vacher et al., 2020*), providing more confirmatory evidence to the geological models of how the basin reached its current configuration.

The main idea in historical biogeography analyses is that geological events have influenced the diversification of the biota, and therefore this diversification pattern is seen in the phylogeny as common cladogenetic patterns, and from a Panbiogeographic point of view, those patterns from different taxonomic groups will agree, following Croizat's idea (*1964*) that the earth and its biota have evolved as a whole. The same idea is found in authors such as *Cunningham & Collin (1994)*, who proposed biogeographic congruence, namely, synchronous vicariant or dispersal events that reflect shared historical processes.

There are biogeographical studies that incorporate a big species sampling from a single group (Frogs or Birds, *e.g.*, *Santos et al., 2009*; *Smith et al., 2014*), but few using species from different taxonomic groups. Also, most of the studies that have addressed the Amazon basin formation have used secondary calibrations (*e.g.*, *Buckner et al., 2015*)—which increase uncertainty in age estimates (*Schenk, 2016*)—or are performed at the population level (*e.g.*, *Ribas et al., 2011*), preventing evaluation of a possible older formation of the Amazon basin. Furthermore, the number of species used has been low in some studies (10–68% less than the number of species used in the present study) (*e.g.*, *Pramuk et al., 2007*; *Maciel et al., 2010*; *Ramírez et al., 2010*). Therefore, the goal of the present study is to evaluate the fit of molecular phylogenetic and biogeographic data to the previously described models (''Old'' and ''Young Amazon'') regarding the age of formation of the Amazon fluvial system, using five hypotheses, varying the dispersal rates on different time scenarios.

## MATERIALS & METHODS

### Selection of taxa

We selected taxa using the following criteria:

(1) We considered only monophyletic taxa with clades currently distributed in the Amazon basin;

(2) The phylogeny must include clades with divergence times covering the temporal ranges proposes by the two models (1.0–11.8 Ma, *Hoorn et al., 2010*; *Figueiredo et al., 2009*; *Figueiredo et al., 2010*; *Ribas et al., 2011*);

(3) At least 60% of the described species in the phylogeny must have both mitochondrial and nuclear genes, and a minimum of three genes available at GenBank;

(4) The taxa must have fossil records available in the literature for the ingroup or the outgroup (see Table S2), and the fossils must not have uncertain phylogenetic positions (*Kay & Meldrum, 1997*; *Kay & Cozzuol, 2006*; *Hosner, Braun & Kimball, 2016*), or poor stratigraphic information.

### Molecular and distributional data

We downloaded all nucleotide gene sequences available at GenBank for each taxon (see Data S1). Each gene was aligned separately using MUSCLE v3.8.31 (*Edgar, 2004*)

with default settings. The nucleotide substitution models were selected with the Akaike Information Criterion (AIC, *Akaike, 1974*) with the modelTest function in the R package 'phangorn' v2.7.1 (*Schliep, 2011*). For the distributional data, we used the available literature and occurrences from the Global Biodiversity Information Facility (http://www.gbif.org/, Table S1). The dataset was curated, removing points out of the known distributional range cited in literature.

## Phylogenetic reconstruction

We carried out a Bayesian partitioned phylogenetic analysis as implemented in MrBayes v3.2.6 (*Ronquist et al., 2012*) *via* the CIPRES Science Gateway (*Miller, Pfeiffer & Schwartz, 2010*). For each partition, we applied the nucleotide model previously selected, and tested whether the evolutionary rate was constant across the phylogeny (strict molecular clock) or it varies on each branch (relaxed molecular clock). Then, we chose the model that best fit our data through Bayes Factors (*Kass & Raftery, 1995*). Once the evolutionary rate of the model was settled, we modelled the relaxed clock using the independent gamma rate model (*Lepage et al., 2007*; *Bakiu, Korro & Santovito, 2015*), which is a continuous uncorrelated model of rate variation across lineages that assumes an independent rate on each branch following a gamma distribution. We used a fossilized birth-death process (*Heath, Huelsenbeck & Stadler, 2014*; *Zhang et al., 2016*) to model the cladogenetic processes, taking into account speciation, extinction and fossilization.

## Priors and clock dating

For each calibration point, we tested two informative prior distributions (exponential and log-normal) and chose the model that best fit the data through Bayes Factors (*Kass & Raftery, 1995*). This step is crucial for the estimation of divergence times of the species. It has been proposed that the reason for over-estimation of branch lengths is the poor choice of the prior distribution (*Rannala, Zhu & Yang, 2011*). Both prior distributions differ in the location of the highest likelihood for the age of a node (*Jones et al., 2013*) and the parameters established for each one (*Ho & Phillips, 2009*; *Arcila et al., 2015*).

We conducted two independent runs with 30 million generations using the selected parameters, sampling every 2,000 generations and discarding the first 25% samples (burn-in). Each run consisted of four Metropolis-coupled Markov Monte Carlo chains with default temperatures ($\Delta = 0.09$ between heated chains). Each analysis was carried out until the average standard deviation of split frequencies was below 0.05 (*Ronquist et al., 2012*). The convergence between runs was assessed using the effective sample size as reported in Tracer v1.7 (*Rambaut et al., 2018*) and the potential scale reduction factor (*Ronquist et al., 2012*). We also used the command 'mcmcp data = no' to perform the analyses without data and sampled from the priors only (*Ronquist et al., 2012*), to evaluate the impact of including only the priors on ages and monophyly.

## Infinite sites

We plotted the posterior means of the divergence of each clade against the 95% posterior confidence interval values (CIs: *Yang & Rannala, 2006*) for each phylogeny to evaluate the uncertainty source in posterior estimates of the divergence times, either by molecular

or fossil sampling. The plot is based on the fact that even with infinitely long sequences, uncertainties will remain in the posterior time estimates because the posterior converges to a one-dimensional distribution (*Yang & Rannala, 2006*).

## Ancestral area inference

We reconstructed the ancestral distribution using areas bounded by major Amazonian rivers (*Alfaro et al., 2015*, Fig. S1), under the dispersal-extinction-cladogenesis model (DEC: *Ree et al., 2005*; *Ree & Smith, 2008*) as implemented in the R (*R Core Team, 2020*) package 'BioGeoBEARS' v1.1.1 (*Matzke, 2018*).

We assessed five different biogeographic hypotheses, the base hypothesis (H0) with a constant dispersal rate in which all areas have the same probability, and four stratified hypotheses with different dispersal rates over time (*Ree & Smith, 2008*), abruptly changing the dispersal rates along the time span, therefore, we can evaluate whether there are changes related to the time and distribution of the ancestral areas in a given temporal range, and if those changes are related to the ranges proposed for the formation of the Amazon basin: (1) The ''Young Amazon'' hypothesis (H1), in which the dispersal rates were low before 11.8 Ma (proposed lower limit, code as 0.25), then started to increase (code as 0.50) during the temporal range in which the Amazon system began forming (11.8–2.5 Ma), and the rates became maximal when the Amazon basin reached its current form and size (after 2.5 Ma, code as 1.0). (2) The ''Old Amazon'' hypothesis (H2), where rates were intermediate (0.50), ranging from 10.0–7.0 Ma, suggesting that the system was mostly formed during this range, and the Amazon basin was completely established after 7.0 Ma (1.0). (3) The Lineage Through Time (LTT) hypothesis (H3, Fig. S2), based on a lineage-through-time plot for all taxa together, and assumed the rates were low until 17.0 Ma (0.25) and became intermediate (0.50) where the PMWS existed between 17.0–10.0 Ma, then, for the maximal rates (1.0) in the last 10.0 Ma to the present, considering that the PMWS drained, and the Amazon basin reached its present form. (4) The River hypothesis (H4), where the rates were minimum until 10 Ma (0.25), intermediate from 10.0–7.0 Ma (0.50) when the Amazon River flowed eastward, and the rates were minimum after 7.0 Ma (0.25) when the Amazon basin was completed.

We used a log-likelihood difference of two units to compare the models (*Edwards, 1992*), and therefore define the best hypothesis.

Additionally, we used RASE v0.3-2 (*Quintero et al., 2015*) as a comparative and confirmatory analysis of BioGeoBEARS. We used the previously dated phylogeny, and transformed the occurrences into polygons using a Minimum Convex Polygon with the R package 'adehabitatHR' v0.4.19 (*Calenge, 2020*). We ran RASE with 50,000 iterations and logged every 10 iterations. As RASE does not estimate ancestral areas, we focused on the direct visual congruence of the dispersal/expansion events between RASE and BioGeoBEARS.

## Isolation barriers inference

We determined the potential isolation barriers for each clade following *Hovenkamp (1997)* and *Hovenkamp (2001)*, who suggested that the only evidence of a speciation process in

a geographical context is the allopatric distributions. We used the Vicariance Inference Program (VIP: *Arias, Szumik & Goloboff, 2011*) and the Geographically explicit Event Model (GEM: *Arias, 2017*) for this analysis. These methods do not require predefined areas, so the percentage of vicariant events can be higher compared with other methods. For VIP, we used a grid sample of 0.25° to minimize undersampling and low resolution, and we used a strict non-overlapping rule. For GEM, following a brief cost exploration, we settled the costs to 1 to the vicariance, and 1,000 to the remaining events to maximize the number of vicariant events.

## Congruence between events

To quantify the congruence, we relied on the approach presented by *Cunningham & Collin (1994)*. The use of different taxa reduces the possible bias that a single taxon could exhibit. We obtained the vicariant events associated with the formation of Amazonian rivers for each clade based on the position of the rivers and through the reconstruction of the ancestral distribution and the evaluation of the vicariant barriers based on grids, and therefore, we did not require prior assumptions of areas. We checked whether the same river could be assigned as a vicariant barrier for two or more taxa, while they shared similar divergence times for that event. The congruence was evaluated for each method separately. We evaluated whether our results are in concordance with the dates found by other biogeographic studies, and dates proposed by geological studies. Finally, we verified whether the two aforementioned methodological approaches showed potential congruent vicariant events associated with the Amazonian rivers.

A methodological workflow can be found in the supplemental information (Fig. S3). Also, all data and R-scripts implemented in this research, and the results are available at: https://github.com/karen9/Amazonia.

## RESULTS

### Taxa and calibration points

The dataset was composed of six phylogenies representing four different taxonomic groups (Table 1). For most of the taxa analyzed in this study, there are no biogeographic studies evaluating the role of rivers as speciation barriers. However, for Cebidae, there are analyses regarding the Madeira (*Buckner et al., 2015*), Negro, Branco, Tapajos, Xingu and Tocantins rivers (*Alfaro et al., 2015*; *Boubli et al., 2015*). For mammals, we found fossil information for the ingroup, while for the remaining groups, we used the fossils that were available for the outgroup (Table S2).

### Molecular clock

The strict molecular clock model was rejected for all genes (Table S3), except for the ATP7A and COI for Stenodermatinae (log-difference of three to five, *Kass & Raftery, 1995*). We verified the individual topologies of each of the two genes to assess the overall impact of these genes, which fit a strict model based on the total evidence topology. We found that they produced unresolved topologies, therefore COI did not present common nodes and ATP7A presented 2% common nodes with the total evidence topology. Nonetheless,
**Table 1 Groups used in the phylogenetic reconstructions.** The number of species used for each group, the percentage of those species that ocurrs in the Amazon basin and the calibration points used.

|  | Species in the phylogeny | Distributed in the Amazon basin | Calibration points |
|---|---|---|---|
| *Rhinella* Fitzinger, 1826 | 41 | 37% | 2 |
| *Melipona* Illiger, 1806 | 39 | 61% | 4 |
| Cracidae Vigors, 1825 | 47 | 53% | 5 |
| Cebidae Bonaparte, 1831 | 44 | 59% | 6 |
| Echimyidae Gray, 1825 | 58 | 43% | 4 |
| Stenodermatinae Gervais, 1856 | 78 | 49% | 3 |

removing the ATP7A and COI genes was not useful, as the removal resulted in a less resolved topology (50% of common nodes) with lower Posterior Probabilities (hereafter, PP).

## Prior and posterior estimates

The exponential and log-normal distributions generated the same topology, but differed in branch lengths. All calibration points for each phylogeny fit best to a log-normal distribution, showing the narrowest 95% CIs without divergence date estimates (Fig. 1). Furthermore, the majority of the calibration points overlapped between the prior and the posterior distributions. For 16 calibration points, the prior overlapped with the posterior about 80–100%, and for nine of these 16 calibration points, the posterior was narrower (Fig. 2).

The only exception in which the prior and the posterior did not overlap was *Pampamys emmonsae* Verzi, Vucetich & Montalvo, 1995 (Echimyidae), where the posterior distribution was older with a difference of 3.0 Ma (median: 17.58, 95% CI [21.58–14.09 Ma] Figure 2). The absence of this fossil in the analysis led to a wider 95% CI (median: 18.84, 95% CI [23.35–14.68 Ma]) than the analysis including it, with a difference of *circa* 1 Million years. However, the PP did not change between both analyses (100%). On the other hand, the incorporation of the new fossil record in the *Cebuella* Gray, 1870 lineage (Cebidae) led to a narrower 95% CI [10.0–10.9 Ma] and a higher PP (100%) than the analysis without this calibration ocurrent point (95% CI [8.7–4.5 Ma], and PP: 73%).

Both, the posterior age means and the CIs fit best to a straight line under the exponential fossil prior distribution (gray, Fig. 1), and led to older ages and a larger 95% CIs than the log-normal distribution (red, Fig. 1). The taxa with the oldest ages and the widest CIs (lower precision) in decreasing order were *Melipona*, Cracidae and *Rhinella*. The taxon with the most recent age and highest precision was Cebidae (Fig. 2).

Finally, the analyses using only priors resulted in unresolved topologies (results not shown), although monophyly was enforced on the calibrated nodes (nodes constrained by fossil ages).

## Ancestral reconstruction

Dispersal is the main driver of most cases of speciation (Fig. 3), and the events reconstructed using BioGeoBEARS are concordant with dates proposed by (*Ribas et al., 2011*). Although

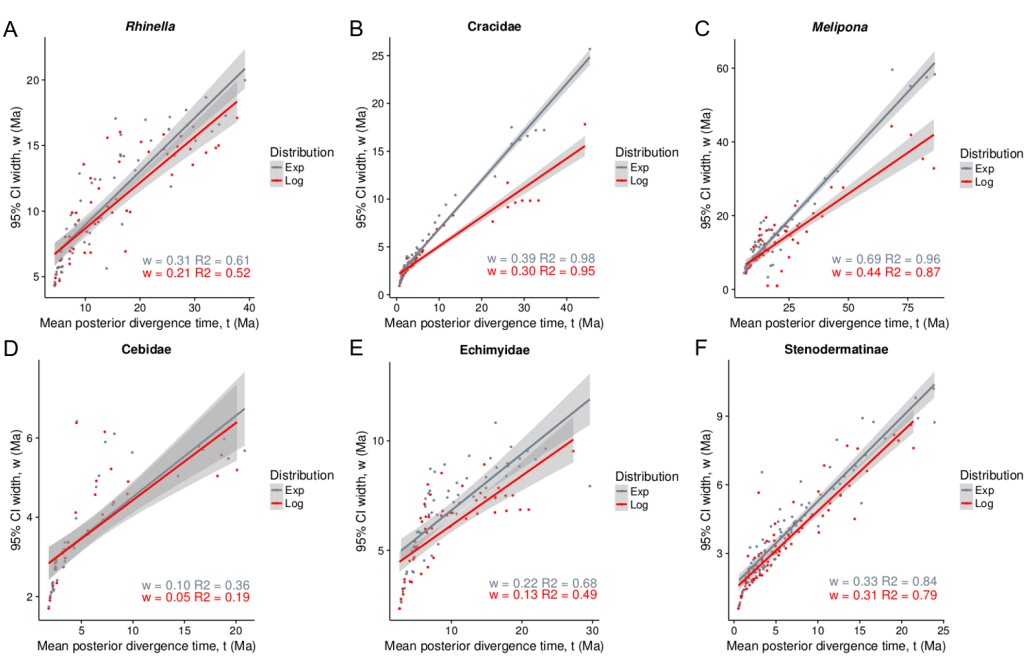

**Figure 1 Infinite-sites plot for all taxa under lognormal and exponential distributions (A–F).** The *x*-axis is the posterior means of each node age, and the *y*-axis is the 95% posterior confidence interval (CI) width values. The slope (w) is a measure of fossil precision and represents the direct relationship between divergence time and uncertainty in the posterior CI (*Yang & Rannala, 2006*).

all taxa fit stratified models, four were better fitted to more than one stratified model, with a log-likelihood difference of 0.1–0.77 (Table S4). Cracidae and Stenodermatinae were the only two taxa that fit the "Young Amazon" model, while the rest of the taxa fit the "Old Amazon" model (Fig. 3). For these two taxa, the first dispersal to the Amazon basin occurred from the Andes, in a temporal range of ~3.0 Ma (95% CI: 0.99) for Cracidae, and ~7.0 Ma (95% CI: 3.3) for Stenodermatinae (Fig. 3). On the other hand, for *Rhinella*, Echimyidae and Cebidae, the first dispersal to the Amazon basin occurred from the Atlantic Forest to some areas of the Western Amazonia (Napo, Marañon, and Ucayali) in the range of ~16.2–10.4 Ma (Fig. 3). Nevertheless, for Cebidae and Echimyidae, the first dispersal involved all areas of the Amazon basin, then experienced a contraction of the ancestral range to the Atlantic Forest and the Andes, but there was no congruence in the temporal range in which these events took place (Fig. 3). Finally, *Melipona* presented both, the Andes and the Atlantic Forest as ancestral area, specifically, has one clade from the Andes and another from the Atlantic Forest, however, the dispersions through the Amazon basin for both clades occurred about ~15.0–9.0 Ma (Fig. 3), fitting the "Old Amazon" model.

With RASE we found similar results to those of BioGeoBEARS, nevertheless given the approach of RASE, we were not able to find the vicariant and contraction events, found with BioGeoBEARS, VIP and GEM (see below). Here, both Cracidae and Stenodermatinae started their dispersion from the Andes through the Amazon basin about ~5.0–3.0 Ma (Fig. 4). For *Rhinella*, Echimyidae and Cebidae the dispersion started *circa* ~10.0 Ma being

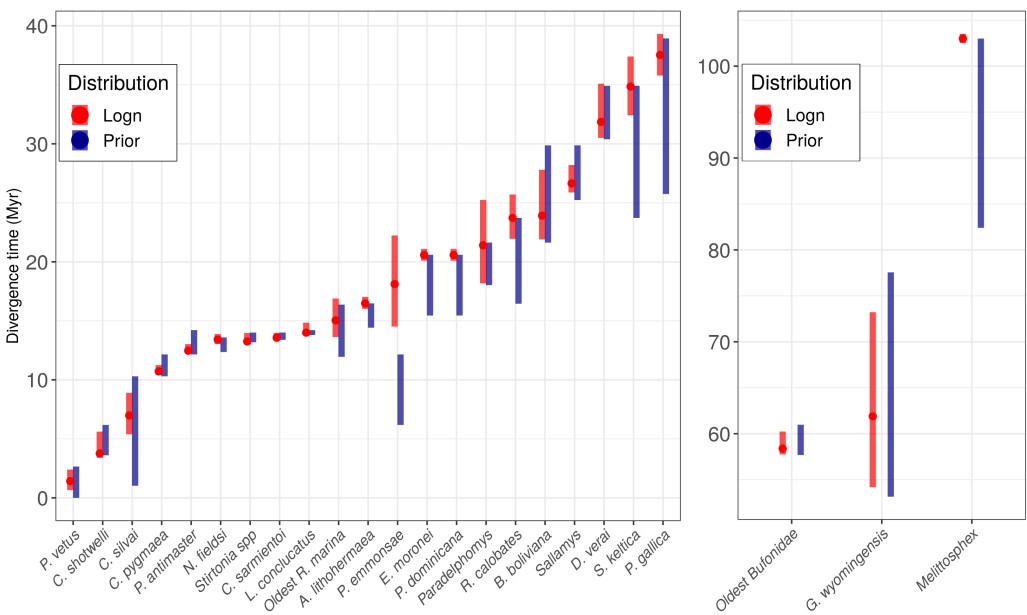

**Figure 2  Prior age distributions and posterior age estimation for all the fossils used.** Red bars indicate the 95% posterior Confidence Interval (CI) under a lognormal distribution, and the points represent the posterior median values. Blue bars correspond to the temporal range for each prior.

more notorious at ∼7.0 Ma, and here, all dispersions started from the southern limit of the Amazonia basin (Fig. 4). For *Melipona* the dispersion dates were similar to those of *Rhinella*, Echimyidae and Cebidae, but they started from the northwestern part of the Amazon basin and the Andes (Fig. 4).

## Isolation barriers and events

Both GEM and VIP recover more vicariant events than BioGeoBEARS, however, we only focused and reported those events that were congruent with BioGeoBEARS and could be associated with a river. We obtained nine vicariant events that might be associated with rivers (Fig. 5). From these events, three were congruent between GEM and BioGeoBEARS, and two between VIP and BioGeoBEARS, while the other four were congruent among the three methods (Fig. 5). Only the Madeira and Tocantins rivers were found as barriers from different taxa in similar time ranges. The Madeira River acted as a barrier for Cebidae and *Melipona* (Figs. 5A, 5F) within ∼8.0–7.0 Ma. On the other hand, the Tocantins River acted as a barrier to Cebidae and Echimyidae (Figs. 5C, 5E) within the range ∼8.0–7.0 Ma. Finally, we found the Amazon River as a possible barrier for three taxa: Cebidae, Cracidae, and *Melipona* (Fig. 5). Nevertheless, the dates of these events are not congruent (∼5, 2, and 10 Ma, respectively).

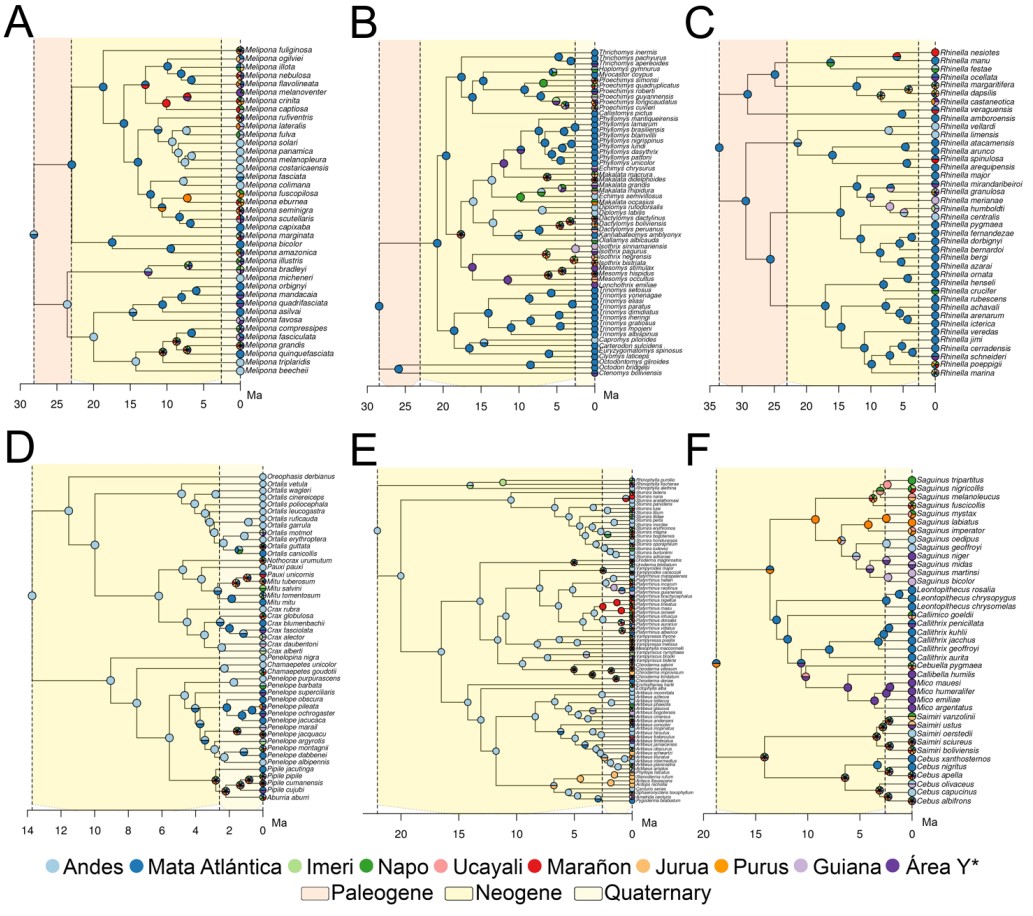

**Figure 3** **Ancestral Range Reconstruction under the model with the smallest likelihood value.**
*Melipona* (A), Echimyidae (B), *Rhinella* (C), Cracidae (D), Stenodermatinae (E), and Cebidae (F). For
Cracidae, Stenodermatinae, and Echimyidae the model has intermediate dispersal rates between 11.8–2.5
Ma (Million years ago) and maximal after 2.5 Ma, and for the rest of the taxa the non-stratified model.
Area Y* = Araguaia+Xingu+Tocantins+Rondônia.

## DISCUSSION

### Historical Amazon

We found that species dispersal agrees with a progressively developing Amazon drainage
system from 16.2–10.4 Ma. We agree with *Jaramillo et al. (2017)* and recent literature (*e.g.*,
*Antoine et al., 2016*; *Hoorn et al., 2017*) that the PMWS was greatly reduced in the middle
late Miocene. Moreover, we propose that the biota follow the development of the Amazon
drainage basin, which, given the data, likely reached its shape and size from ~12.2–6.2 Ma
(Figs. 6C–6D), when the taxa extended their distributions to Eastern Amazonia. In general,
the temporal range found here agrees with the "Old Amazon" model, but we might
consider a wider temporal range than those proposed by *Hoorn et al. (2010)* and *Hoorn et
al. (2017)*.

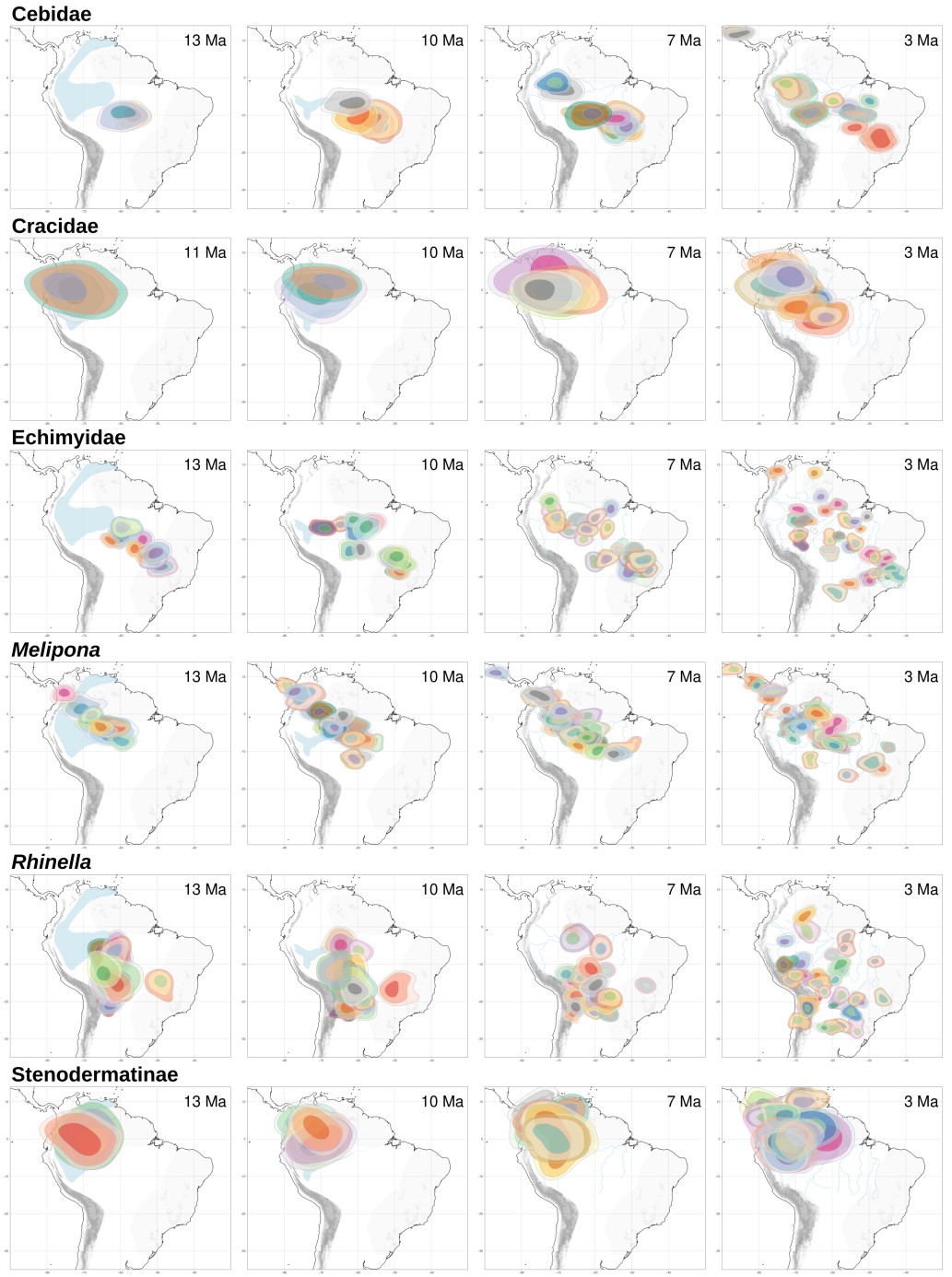

**Figure 4** **RASE results.** Ancestral ranges for the six studied taxa at four time slices of interest for the "Old Amazon" and "Young Amazon" models. In blue, the PMWS through time, according to *Hoorn et al. (2010)* and *Jaramillo et al. (2017)*.

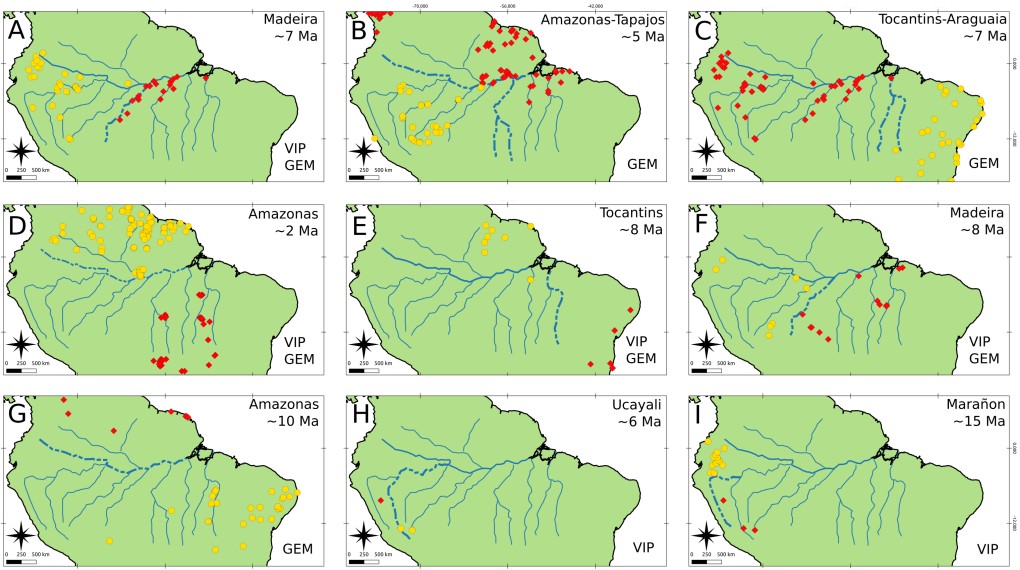

**Figure 5** **Isolation barriers and events.** Rivers found to be, probably, biogeographical barriers (dashed line) as determined under Hovenkamps's context (1997, 2001) for the following taxa: (A–C) Cebidae, (D) Cracidae, (E) Echimyidae, (F–G) *Melipona* and (H–I) *Rhinella*. All events were found with BioGeoBEARS, and also it is stated if the event was recovered by GEM, VIP or both.

Although both Cracidae and Stenodermatinae fit a stratified model of the "Young Amazon" hypothesis where the dispersal rates are maximum after 2.5 Ma, the reconstruction of the ancestral range indicates a different scenario in which the first dispersal to the Amazon basin occurred before 2.5 Ma, and occurred at different ranges for each taxon. However, we cannot discard the possibility that after the first dispersal to the basin, the number of dispersal events could have increased in response to the complete establishment of this fluvial system.

The temporal range found here for the formation of the Amazon fluvial system agrees with the geological process sequence that occurred during that time. *Hoorn et al. (1995)* and *Hoorn et al. (2010)* suggested that the formation of the Amazon River can be attributed to the Andean uplift, and our findings are consistent with stratigraphic studies regarding the emergence of the Eastern Cordillera of the Colombian Andes (*Mora et al., 2008*), and with paleobotanic observations, which suggest that elevations of the Eastern Colombian Andes were homogeneous between the early and middle Miocene and Pliocene (*Gregory-Wodzicki, 2000*), as well as with studies regarding the sedimentation rates of the Foz do Amazonas, which have highly increased since the middle Miocene (*Figueiredo et al., 2009*; *Figueiredo et al., 2010*; *Hoorn et al., 2017*). On the other hand, our results are also congruent with other geological events, such as the date of activation of the Vaupés Arch in the Miocene (*Jaramillo et al., 2017*) and the subsidence of the Purus arch that allowed the Amazon to flow to the east. Although in the present study we can not establish the exact range of its subsidence, we can say that it might be assigned to the middle Miocene, which contrasts with other proposals (*e.g., Nogueira, Silveira & Guimarães, 2013*). The geological events

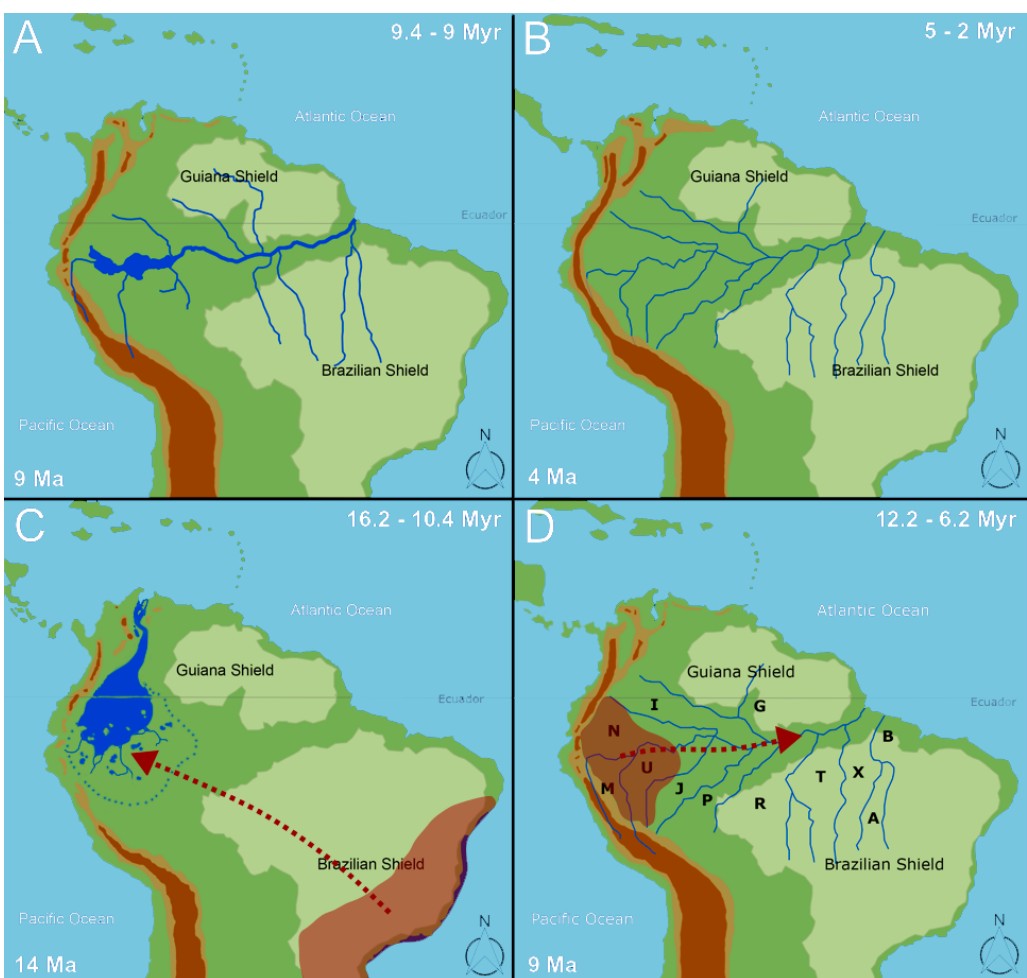

**Figure 6 The biogeographical pattern for formation of the Amazon basin.** Maps are modified from *Hoorn et al. (2010)*. Dates for the closure of the Panama isthmus according to *Bacon et al. (2015)*, Caribbean islands according to *Ali (2012)*. The numbers in the bottom right correspond to Andean uplift according to *Hoorn et al. (2010)*. The panels (A) and (B) correspond to the temporal events proposed by *Hoorn et al. (2017)* for the transcontinental rivers and the establishment of the Amazon fluvial system, respectively. The panels (C) and (D) present our interpretation of the temporal events found in the present study, in which the first dispersals occur to Western Amazonia (Napo, Marañón, and Ucayali) and the expansion of the distributional range from Western Amazonia to Eastern Amazonia. Area labels as follows: Guiana (G), Imeri (I), Rondônia (R), Tapajos (T), Belem (B), Mata Atlantica (F), Xingu (X), Napo (N), Marañon (M), Ucayali (U), Jurua (J), Purus (P), Araguaia (A), Andes (D), Y Area = Araguaia+Xingu+Tocantins+Rondônia. Panel C is only a graphical representation of the retraction of PEBAS and the formation of the Amazon rivers.

together have been broadly accepted as the main causes of the disappearance of the PMWS. Notwithstanding, our data and analyses do not allow us to establish the complete temporal range in which the system existed.

Our results are not affected by the number of time slices and different probabilities of dispersal in the range-inheritance scenarios (*Ree & Smith, 2008*), and consequently, we cannot establish the temporal ranges in which there is a change in the dispersal rates

according to the geological ranges proposed for the two aforementioned hypotheses. Moreover, using the biogeographical approaches we demonstrated that the large Amazonian rivers can limit species distribution ranges (*Moraes et al., 2016*; *Oliveira, Vasconcelos & Santos, 2017*), while the dates found here differ from *Ribas et al. (2011)* and *Alfaro et al. (2015)* -which found events associated with the Tocantins river about 0.8–0.3 Ma, and *Buckner et al. (2015)*, which found events for the Madeira river about 5 Ma-. It is worth noting that the methods used in this study to identify rivers as vicariant barriers are based on different algorithms, and therefore, different results could be expected. The main differences between our findings and those of other authors who tested or evaluated rivers as vicariant barriers are the type of analyses and the implemented data. Some studies have incorporated populations and species (*Ribas et al., 2011*; *Buckner et al., 2015*), while some only took into account mitochondrial genes (*Alfaro et al., 2015*) or secondary calibrations (*Alfaro et al., 2015*; *Buckner et al., 2015*) and did not carry out an exploration of priors.

During the last 20 Ma, the climate experienced drastic and rapid changes, ranging from warm to cool climate (*Flower & Kennett, 1993*; *Holbourn et al., 2015*; *Westerhold et al., 2020*). These drastic changes directly affect the ecosystem, especially the plant community and may be a factor for the actual species distributions, especially for the dispersion event. Although climate and its effect on species distributions is clear, we do not have a compressive database to our time range to test the effects of climate on the dispersal events.

## Ages estimates

The chosen priors are reasonable for modeling our data as there is a considerable overlap between the prior and the posterior density functions, with the posterior more concentrated than the prior (Fig. 2, *Nascimento, Reis & Yang, 2017*), but priors only do not determine the results. The infinite-sites plot (Fig. 1) suggests that the uncertainty in the posterior age estimates is mainly due to limited molecular data (*Yang & Rannala, 2006*) and is not due to the fossil points used, as is reflected in the regression value, which is low in most cases (except for Cracidae).

We cannot rule out the possibility that both an increased density of taxon sampling at genome scale and fossil sampling could improve age estimates (*Yang & Rannala, 2006*). We noted that incorporation of fossils for the clade in which posterior is older and out of the prior (*e.g.*, *Pampamys emmonsae*) led to more precise age estimation (*Foote et al., 1999*; *Smith & Peterson, 2002*; *Sytsma, Spalink & Berger, 2014*).

The molecular clock model used here does not incorporate uneven fossil sampling (*Drummond et al., 2012*; *Zhang et al., 2016*), which could have impacted the posterior age estimates. Thus, the ancient and less precise ages produced in the present study could be due to a lack of internal node constraints, which leads to ancient ages (*Bibi, 2013*; *Arcila et al., 2015*). This could be highlighted by the fact that the taxon with most fossils (Cebidae) presents more precise date estimates for the entire dataset, and the amount of uncertainty added in the posterior CI is lower (Fig. 1). On the other hand, in contrast with *Heads (2012)*, our results show that the use of an exponential distribution for the calibration points generates older and less precise posterior estimates (Fig. 1), as was previously suggested (*Heath, 2012*; *Sauquet et al., 2012*; *Arcila et al., 2015*).

## CONCLUSIONS

Our results provide support for the "Old Amazon" hypothesis, as well as for a middle late Miocene time origin for the Amazon drainage system. We only obtained evidence for the date of formation of two rivers (Tocantins and Madeira), although there might be other rivers acting as biogeographical barriers. For the temporal range studied here, the rivers did not structure the Amazonian biota. It is likely that there are other physical factors involved in Amazonian biota evolution, emphasizing the complexity and dynamics of the Amazonian system. It is, therefore, necessary to consider this issue with different tools with multiple sources of data. Furthermore, our analyses highlight the importance of including numerous fossil calibration points, distributed throughout the phylogeny, and an exploration of priors, resulting in more precise age estimates.

## ACKNOWLEDGEMENTS

We thank Juan David Bayona-Serrano and Yelsin Méndez-Camacho, for their help preparing the initial draft of the manuscript, to Daniel Pabón for advice and help in the handling of supercomputing platforms, and all the members of the Laboratorio de Sistemática y Biogeografía who contributed with feedback. We are grateful to Carina Hoorn and Christine Meynard, which greatly improved the final version. Furthermore, we thank the CIPRES facility for their available computational resources.

### Funding

Daniel R. Miranda-Esquivel was supported by projects 8867: "Inventario de la diversidad biológica en una región del sur de Bolívar, Colombia", and 8034: "Una expedición para reducir el déficit de conocimiento en biodiversidad a una escala en Santander, Colombia". MinCiencias-Colombia/Vicerectoría de Investigaciones UIS. The funders had no role in study design, data collection and analysis, decision to publish, or preparation of the manuscript.

### Grant Disclosures

The following grant information was disclosed by the authors:
8867: "Inventario de la diversidad biológica en una región del sur de Bolívar, Colombia".
8034: "Una expedición para reducir el déficit de conocimiento en biodiversidad a una escala en Santander, Colombia".
MinCiencias-Colombia/Vicerectoría de Investigaciones UIS.

### Competing Interests

The authors declare there are no competing interests.

### Author Contributions

- Karen Méndez-Camacho conceived and designed the experiments, performed the experiments, analyzed the data, prepared figures and/or tables, authored or reviewed drafts of the paper, and approved the final draft.

- Omar Leon-Alvarado analyzed the data, prepared figures and/or tables, authored or reviewed drafts of the paper, and approved the final draft.
- Daniel R. Miranda-Esquivel conceived and designed the experiments, analyzed the data, authored or reviewed drafts of the paper, and approved the final draft.

## Data Availability

The data and R-scripts are available at GitHub: https://github.com/karen9/Amazonia.

## Supplemental Information

Supplemental information for this article can be found online at http://dx.doi.org/10.7717/peerj.12533#supplemental-information.

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
