# Peer review of "Biogeographic evidence supports the Old Amazon hypothesis for the formation of the Amazon fluvial system"

_PeerJ, doi:10.7717/peerj.12533_

## Round 0.1 · original submission · Major Revisions

I have received now two very detailed reviews about your manuscript. Both reviewers agreed that this is a quite interesting paper with a potential impact on the field. I agree with them. However, they also pointed out some problems with interpretations of results of biogeographical analysis. R2 also mentioned that you need to actually compare the taxa in a more quantitative way. R1 provided a number of references that are missing, including recent ones, that definitely need to be incorporated in the introduction and discussion. I fully agree with him that there are more recent geological/paleontological papers on the formation of the Amazon basin that you didn't cite. Please, pay close attention to those comments.

In Fig 1, prefer to use Atlantic Forest, as appears in the rest of the text.

I highly recommend you to follow the structured abstract of PeerJ when re-submitting the revised version of the text.

I also highly recommend you to structure the end of the introduction as a series of questions and respective hypotheses.

R1 made some comments in your Table S2, which I'm attaching here.

·

Basic reporting

This manuscript uses the spatiotemporal pattern of six Metazoan clades in order to test the Old vs. Young Amazon hypothesis. This is to be commended, as this question has been hotly debated in the last decades.

In my opinion, one of the main issues is the relevance of such six clades for solving this decade-long controversy. I may be wrong, but I suspect that way more clades may follow all four criteria as summarized in the “Selection of taxa” subsection (see 101-109) among the myriad of Metazoan genera and families documented in the Amazon basin. Apart from these criteria, nothing is said about the significance of such sample for addressing the question, or the potential biases it may suffer from. Moreover, I see no clear consistency in this sample: the authors have studied a bee genus (volant and social), an amphibian (non-volant / semi-aquatic), as well as a bird family (volant) and three mammalian families (arboreal monkeys; rodents with a wide array of locomotory and habitat preferences; phyllostomid bats, volant and mostly fruit-eating). In other words, these groups have very distinct biogeographical histories, feeding habits, and dispersal/vicariant behaviors and capabilities when facing potential biogeographical barriers (especially overwater).

“Most studies support a “Young Amazon” model” (line 81). I do not agree, as the trend had considerably changed in the last decade. The literature survey as provided in the introduction fully ignores an entire range of recent references. Sedimentologists, structural geologists, palynologists, and vertebrate paleontologists have gathered a bunch of data supporting the decay of the PMWS and the onset of a transcontinental Amazon drainage by early late Miocene times in the last decade. See for instance Roddaz et al. (2010; Hoorn & Wesselingh book), Boonstra et al. (2015, Palaeo3), Salas-Gismondi et al. (2015, Proc Roy Soc B; 2016 PLoS ONE), Antoine et al. (2016; Gondwana Research), Jaramillo et al. (2017, Science Advances), Marivaux et al. (2020, J Hum Evol). Moreover, some studies have demonstrated that the Amazon basin had been split into two major subbasins (South vs. North) no earlier than in latest Miocene to Pliocene times (e.g., Espurt et al., 2007 Geology), which seems to match the biogeographical history of various plant and animal clades from Amazonia. In other words, the Amazonian geological history is much more complicated than the authors seem to depict it, in that events involving Amazon River and its tributaries have obviously occurred throughout the last 12 My.

Similarly, the authors write that they “agree with Shephard (2010), who suggested that the PMWS was greatly reduced in the middle-late Miocene.” This is fully at odds with more recent literature providing primary data on palynomorphs, marine microfossils, and aquatic communities (Boonstra et al., 2015, Palaeo3; Antoine et al., 2016, Gondwana Res; Jaramillo et al., 2017, Sci Adv; Feussom Tcheumeleu et al. (2019, Palaeodiv & Palaeoenv). In this scheme, an updated map is available for middle Miocene times (PMWS: see map in Marivaux et al., 2020, J Human Evol). It is adapted from that of Boonstra et al. (2015, Palaeo3).

To end with, a similar study focused on biogeographical history of Amazonian frogs was published by Vacher et al. (2020, J Biogeogr). It somewhat supports the current results, with a distinct geographical grid and pattern. It may be worth mentioning it.

Experimental design

As a paleontologist, I have not addressed molecular phylogeny-based methodology. Instead, I have focused my revision on geological premises, paleontological literature and taxonomic issues, notably related to the fossil record, in order to improve the manuscript.

Validity of the findings

Aside from the geological context, the authors seem to have missed some recent literature of primary interest regarding calibration points, at least for Echimyidae (Boivin et al., 2019 Geodiversitas; Courcelle et al., 2019, Mol Phyl Evol; Pérez et al., 2019, J Syst Palaeont) and Cebidae. For instance, the cebine Panamacebus is dated at c. 21 My (Bloch et al., 2016, Nature) and Cebus sp. was recognized in c. 11.5 My-old deposits (Marivaux et al., 2016 AJPA [cited for Cebuella, but seemingly not used for Cebus].
This may affect the ages of several dichotomies in the Fig. 4 and probably reinforce the authors’ conclusions. Accordingly, the Supplementary table with taxa and their ages needs to be widely enhanced, as some references are outdated, systematic assignment of several taxa is questionable, and more precision would be required. For instance, “Middle Miocene” is not precise enough as an age for cebids found at La Venta (see primary references in Kay, Madden, Cifelli, and Flynn, 1997; La Venta book). I would also recommend citing updated primary references (note that no reference section is available in the SI).

Moreover, the periods ranging the 10–6 My and 2–0.5 My time intervals coincide with drastic global climatic changes (e.g., Westerhold et al., 2020, Science), mainly affecting rainfall in tropical-equatorial areas (Martínez et al., 2020, Sci Adv). At a regional or a local scale, this has in turn affected plant community composition, with cascading consequences on animals feeding on plants and those feeding on insects (i.e., all clades studied here). In other words, the authors should be aware of the existence of confounding factors. I would suggest to clearly mention such caveats.

Additional comments

I have uploaded an annotated pdf of the main manuscript with some typos and minor issues, remarks, and suggestions. I could not upload the annotated version of the Supplementary data I had prepared (using track changes), as only one file can be uploaded on the review page but I keep it available if needed.

·

Basic reporting

The manuscript uses clear English, but a few statements are somewhat ambiguous. I comment on these and other issues in my line-by-line. No specific hypotheses are put forward – I believe the authors have the necessary background information and are working on an appropriate system to do so

Experimental design

The manuscript uses clear English, but a few statements are somewhat ambiguous. I comment on these and other issues in my line-by-line. No specific hypotheses are put forward – I believe the authors have the necessary background information and are working on an appropriate system to do so

Validity of the findings

The results are very interesting for our understanding of Amazonian biogeography. However, I believe some assertions/conclusions should be alleviated, in particular the one that already appears in the abstract and is developed in the discussion – “we suggest that the temporal range for the beginning and complete formation of the Amazon fluvial system might be older than previously proposed”. The method the authors used to estimate phylogenies and associated diversification times is robust, however, there are theoretical principles that are not taken into account when such conclusions are put forward (please check my main concern #1 in the line-by-line review). Both fossils and trustworthy biogeographic information could be used to estimate divergence times. What the authors are implying is that the former would provide more realistic information on the actual divergence times and, hence, could be used to assess our current knowledge on the geologic dating of the formation of such geologic events (the formation of the rivers) and its importance to understand the biogeography of the region. The arguments on lines 312-314 highlight, again, my main concern #1: your analyses cannot be used to ‘assign’ dates to geologic events such as the subsidence of the Purus. They can only provide evidence that these dates might need to be reassessed, or that the particular taxa used in this study do not show biogeographic patterns that support the currently recognized dates (but others might).

Additional comments

Please find my complete line-by-line review below. I raise three points that I consider of particular relevance, and that should be addressed when preparing a revised version of the manuscript.

1) Dated geologic events are the main information that evolutionary biologists can use to investigate the history of life on earth, and molecular dating evidence can be used to corroborate the geologic evidence. However, the opposite is not necessarily true unless we are talking about poorly understood system. Much more needs to be understood about the geologic history of the Amazon basin, but it is certainly not a poorly understood system (Wesselingh et al., 2011). The fact that in many instances there is an inversion of this interpretation should definitely be addressed by the authors (e.g., my comments on line 61–63).
2) The lack of a quantitative comparative approach, instead of a visual comparison of results.
3) The lack of a clearly-stated hypothesis-testing framework. One can infer the author’s working hypothesis, but clearly stating it would make the presentation of results easier to follow.

57: typo, should be “Plio” not “Pilo”.

59: Ribas et al 2011 should be 2012.

61-63: This assertion relates to my main concern on this study – the interpretation that using dated phylogenies can provide estimates for the dating of the geologic events, which seems to be an inversion of how the evidence should be used in biogeographic studies. Of course, biogeographic information can and should be used to corroborate the estimates made from geologic information. However, geologic dating is the main primary evidence for the history of the planet and not the opposite. Robust geologic dating cannot simply be refuted using molecular dating techniques, since those are known to have many caveats (Duchene et al., 2014; Bromham et al., 2017). Also, even assuming that a molecular dated phylogeny provides ‘perfect’ dates, the processes behind the actual dates might not be the geologic events in question. Many other evolutionary/biologic processes can be invoked to explain the evolution of particular clades in a particular biogeographic scenario. The null hypothesis that a simple vicariant biogeographic scenario is the best evolutionary scenario will certainly not hold true in many cases. The processes behind the divergence times must be interpreted in a population-based scenario, i.e., other evolutionary mechanisms that, for example, cause changes in population sizes, or introgression among the selected clades, might influence the estimation of branch lengths and, hence, dating results in a Bayesian framework (Ho & Duchene, 2014; Bromham et al., 2017; Dos Reis et al., 2018), and not only major biogeographic events. In summary, Alfaro et al 2015 do not, in any way, propose plausible dates for the formation of Amazonian rivers – they do provide evidence that the evolutionary history of Saimiri was influenced by the rivers, supporting a geologic model that was built based on geologic (and not molecular dating) evidence.

77: “either” implies that another reason would also be put forward, which was not the case.

81: Should be a comma after “general”, not a period.

81–83: Background knowledge on Amazonia biogeography is only briefly reported, although I understand the approach considering the vastness of the subject. However, I believe it would be interesting to develop a better argument on the possible reasons why some studies support a “young Amazon” model and others an “old amazon”. In the next paragraph, a few reasons are put forward (secondary calibrations, number of species, studies at the population level, etc), but they are only mentioned as possible problems. Considering the scope of this manuscript, I believe that a paragraph dedicated to highlighting the main differences between studies that corroborate young versus old models would be highly beneficial to the readers, and also to support the aims of the study.

85–89: Although I appreciate the brief reminder on historical biogeography and panbiogeography, it seems that appropriate references are missing (Croizat is actually cited in the reference list, but no publication date appears here).

89: You mean that there are no “biogeographical studies that incorporate different taxa” for the study of Amazonian diversification? This is a strong assertion, and I highly disagree with it. First, what do you mean by taxa here? You used different taxonomic ranks here, which is fine, but other studies have certainly incorporated much more dense sampling, covering dates that could certainly indicate both old and new diversification, and used many taxa as well, such as (Santos et al., 2009; Smith et al., 2014).

95–97: “the goal of the present study is to evaluate the fit of molecular phylogenetic and biogeographic data to the previously described models regarding the age of formation of the Amazon fluvial system”. The way that this is put forward is correct and contradicts the way other arguments are built in the study (see my main concern #1).

Material and Methods
I commend the authors for their clear and well-designed selection of taxa strategy.

I am also particularly happy with the phylogenetic reconstruction and divergence times estimation methods. The authors employed the best available methods for their case and used many strongly recommended error-checking strategies, and I can only congratulate them for that.

159–165: I am not convinced that the VIP analysis was suitable for this case. It seems to be a highly simplistic model and, more importantly, is not able to actually provide the information that is presented in Fig. 5. I understand one can assume that the highlighted river was the biogeographic barrier responsible for the vicariant event, but this is not actually tested (and only visually inferred as far as I can tell). There is a more modern version of this type of analysis, written and implemented by the same author (Arias, 2017). Nonetheless, I suggest the authors to use a model-based inference based on evolutionarily explicit models - rase (Quintero et al., 2015), in line with their beautifully performed phylogenetic inferences.

The BioGeoBEARS analyses were also quite nicely performed, thank you.

Congruence between events (main concern #2): it seems to be a problem in the proposed “comparative” approach since no statistically explicit comparison is performed, but a simple interpretation of congruence based on visual estimates. There are a few available methods that could infer the concordance among the dates (Oaks, 2019) or, maybe more important in this case, estimate the congruence among the vicariant events (nodes on the trees) (Satler & Carstens, 2016).

Results
232–249: A few sections could be improved for clarity.

251–252: The fact that the analyses using only the priors resulted in unresolved topologies is not surprising. Running the analyses only using the priors should be used to assess how informative the data are, and also to check if the priors were correctly specified. This was already reported in the previous paragraphs, so the authors might consider removing this information.

Isolation barriers and ancestral reconstruction: it would be much easier to understand this section if the results were directly associated with the proposed models (young versus old). This relates to my main concern # 3: if a hypothesis-testing framework was put forward, the importance and understanding of each of these results would be largely improved.

Discussion
285: please use ‘proposes’ instead of ‘recreates’.

339: ‘than’ instead of ‘that’.

353: ‘less precise’ in relation to what?

I appreciate how some potential problems with the divergence times estimations are taken into consideration, but I suggest that these are presented in light of their implication; e.g., how would your interpretation of the whole system change considering that your ages might be older because no internal constraints were present for most taxa?

Figure 3: It seems that two distinct plots were used to visually accommodate the differences in the age scale (y-axis). I suggest the authors write this out in the legend.

Figure 4: the authors might consider depicting all trees using the same scale, so they can be directly compared.

References cited in the review:
Arias, J.S. (2017) An event model for phylogenetic biogeography using explicitly geographical ranges. Journal of Biogeography, 44, 2225-2235.
Bromham, L., Duchene, S., Hua, X., Ritchie, A.M., Duchene, D.A. & Ho, S.Y.W. (2017) Bayesian molecular dating: opening up the black box. Biol Rev Camb Philos Soc,
Dos Reis, M., Gunnell, G.F., Barba-Montoya, J., Wilkins, A., Yang, Z. & Yoder, A.D. (2018) Using Phylogenomic Data to Explore the Effects of Relaxed Clocks and Calibration Strategies on Divergence Time Estimation: Primates as a Test Case. Syst Biol,
Duchene, S., Lanfear, R. & Ho, S.Y. (2014) The impact of calibration and clock-model choice on molecular estimates of divergence times. Molecular Phylogenetics and Evolution, 78, 277-89.
Ho, S.Y. & Duchene, S. (2014) Molecular-clock methods for estimating evolutionary rates and timescales. Molecular Ecology, 23, 5947-65.
Oaks, J.R. (2019) Full Bayesian Comparative Phylogeography from Genomic Data. Syst Biol, 68, 371-395.
Quintero, I., Keil, P., Jetz, W. & Crawford, F.W. (2015) Historical Biogeography Using Species Geographical Ranges. Syst Biol, 64, 1059-73.
Santos, J.C., Coloma, L.A., Summers, K., Caldwell, J.P., Ree, R. & Cannatella, D.C. (2009) Amazonian Amphibian Diversity Is Primarily Derived from Late Miocene Andean Lineages. PLoS Biol, 7, 0448–0461.
Satler, J.D. & Carstens, B.C. (2016) Phylogeographic concordance factors quantify phylogeographic congruence among co-distributed species in the Sarracenia alata pitcher plant system. Evolution, 70, 1105-1119.
Smith, B.T., McCormack, J.E., Cuervo, A.M., Hickerson, M.J., Aleixo, A., Cadena, C.D., Perez-Eman, J., Burney, C.W., Xie, X., Harvey, M.G., Faircloth, B.C., Glenn, T.C., Derryberry, E.P., Prejean, J., Fields, S. & Brumfield, R.T. (2014) The drivers of tropical speciation. Nature, 515, 406–409.
Wesselingh, F.P., Hoorn, C., Kroonenberg, S.B., Antonelli, A., Lundberg, J.G., Vonhof, H.B. & Hooghiemstra, H. (2011) On the Origin of Amazonian Landscapes and Biodiversity: A Synthesis. Amazonia: Landscape and Species Evolution (ed. by C. Hoorn and F.P. Wesselingh). Blackwell Publishing Ltd.

---

## Round 0.2 · Minor Revisions

I have now finally heard back from the same two reviewers who commented in the first version. I congratulate the authors for their hard work in addressing each of the reviewers' critiques and submitting a clean version.

I'm recommending minor revision because R1 mentions the lack of Pebas wetland in one of the figures and R2 points that the first paragraph of the introduction can be improved and the phrasing about the phylogenetic inference using only priors can be more clear.

Please, make these final amendments and I believe the manuscript will be accepted afterwards.

·

Basic reporting

This manuscript is the revised version of a work that I had the opportunity to evaluate. This new version is widely improved with respect to the original one, and most of my comments have been taken into account (most premises, rationale, and literature).

Experimental design

Geological premises, paleontological literature and taxonomic issues, notably related to the fossil record, are now correct.

Validity of the findings

The results appear to be robust; the discussion is clear and the conclusions are not oversold.
The only sticking point is the depiction of a fully modern Amazon River network 14 Mya (Fig. 6C), both lacking the Pebas Mega-Wetland System and running continuously between the Andes and the Atlantic Ocean. There are no geological grounds for this. On the contrary, several authors (Croft, 2007 – Palaeontology; Antoine et al., 2013 – op. cit.; Marivaux et al., 2020, J Human Evol) use the concept of “western corridor” to explain the relative uniformity of terrestrial vertebrate components along the central to northern Andes. This could be a good premise.

Additional comments

Some references listed in the text do not follow a chronological/alphabetical sequence (lines 51-53, 60-61, etc.).

Line 64 and throughout the text (e.g., 346, 434): “middle-late Miocene” seems to be used in a wrong way. From what I have understood, the authors mean ‘the middle part of late Miocene times’ (around 10-9 Mya), which should be “middle late Miocene” (i.e., without the minus sign). When using the minus sign, the meaning is fully distinct, in pointing to the complete time interval encompassing middle and late Miocene times. Alternatively, f the authors mean the short time between the middle Miocene and the late Miocene, then they should use the “middle-late Miocene transition”.

Lines 71 and 75: “changed”

Line 86: “Pliocene-Pleistocene”

Line 126: “Meldrum”

Line 227: “undersampling”

Lines 369 and 374: “middle Miocene” instead of “mid-Miocene’

Figure 6: The panel C fully ignores the comprehensive corpus of data documenting the Pebas Mega-Wetland System by 17-10 My (see paleomaps by Antoine et al., 2013 – op. cit.; Boonstra et al., 2015 – Palaeo3; Marivaux et al., 2020 – J Human Evol [about a cebid, 13 My-old!]). In other words, there should be a “mega-pantanal” in Western Amazonia instead of a modern-like Ucayali-Marañon River network. Perhaps, both could be superimposed graphically, to help the readers.
The authors are also suggested to take into account the western Corridor hypothesis (Croft, 2007; Antoine et al., 2007 – 4th EMPSLA proceedings; Marivaux et al., 2020 – J Human Evol) for explaining the expansion of distributional ranges between Atlantic areas and Western Amazonia by that Pebasian-climax time.
Moreover, there is absolutely no argument in favor of an uninterrupted transcontinental drainage (Andes–Atlantic Ocean) at that time and their results do not allow the authors to pass through such geological/geographical grounds. In other words, I would recommend them not to draw a continuous line between Western Amazonia and the lower Paleo-Amazon Basin (the mouth of which was probably located more to the south).

Pierre-Olivier Antoine

·

Basic reporting

The revised version of the manuscript is a much-improved version, and I congratulate the authors for their efforts in responding to all my concerns. The manuscript uses clear English, but a few section could be improved for stylistic reasons – I briefly comment on those below, but please use them at your discretion – they are not mandatory changes, but only suggestions.

Experimental design

The research questions are clear, and the phylogenetic and biogeographic methods are particularly robust in this revised version of the manuscript.

Validity of the findings

The results are very interesting and add important evidence to our understanding of Amazonian biogeography.

Additional comments

The first paragraph consists of only two short sentences, highly contrasting with the second one. The riverine barrier hypothesis was strongly developed since the publication of the seminal work of Haffer on the distribution of lowland vertebrate Areas of Endemism. I understand the authors intention on not particularly touching on this subject, but a few studies do provide the means to incorporate such novel information on the background they choose to develop, such as (Santorelli et al., 2018), and (Oliveira et al., 2017).

I reiterate that only using the priors resulting in unresolved topologies is not surprising – the analyses have no information to rely on in order to obtain branch lengths and bipartitions. These results should be used to compare the fit of the data to the prior (Nascimento et al., 2017). I do believe the authors used this information well, and no actions should be taken apart from, maybe, changing the text (nonetheless, it should be noted that it is expected that the posterior is narrower, and cases where it is not might indicate that the calibration points are not informative, although this should be interpreted with caution).

Nascimento, F.F., Reis, M.d. & Yang, Z. (2017) A biologist’s guide to Bayesian phylogenetic analysis. Nature Ecology & Evolution, 1, 1446-1454.
Oliveira, U., Vasconcelos, M.F. & Santos, A.J. (2017) Biogeography of Amazon birds: rivers limit species composition, but not areas of endemism. Scientific Reports, 7
Santorelli, S., Jr., Magnusson, W.E. & Deus, C.P. (2018) Most species are not limited by an Amazonian river postulated to be a border between endemism areas. Sci Rep, 8, 2294.

---

## Round 0.3 · accepted · Accept

Thank you for submitting these final corrections. I'm glad to recommend the paper as is for publication.